# Review of Thin-Layer Chromatography Tandem with Surface-Enhanced Raman Spectroscopy for Detection of Analytes in Mixture Samples

**DOI:** 10.3390/bios12110937

**Published:** 2022-10-28

**Authors:** Meizhen Zhang, Qian Yu, Jiaqi Guo, Bo Wu, Xianming Kong

**Affiliations:** 1School of Petrochemical Engineering, Liaoning Petrochemical University, Fushun 113001, China; 2Jiangsu Co-Innovation Center for Efficient Processing and Utilization of Forest Resources and Joint International Research Lab of Lignocellulosic Functional Materials, Nanjing Forestry University, Nanjing 210037, China; 3School of Electrical Engineering and Computer Science, Oregon State University, Corvallis, OR 97331, USA

**Keywords:** TLC-SERS, separation, detection, real sample

## Abstract

In the real world, analytes usually exist in complex systems, and this makes direct detection by surface-enhanced Raman spectroscopy (SERS) difficult. Thin layer chromatography tandem with SERS (TLC-SERS) has many advantages in analysis such as separation effect, instant speed, simple process, and low cost. Therefore, the TLC-SERS has great potential for detecting analytes in mixtures without sample pretreatment. The review demonstrates TLC-SERS applications in diverse analytical relevant topics such as environmental pollutants, illegal additives, pesticide residues, toxic ingredients, biological molecules, and chemical substances. Important properties such as stationary phase, separation efficiency, and sensitivity are discussed. In addition, future perspectives for improving the efficiency of TLC-SERS in real sample detecting are outlined.

## 1. Introduction

Surface-enhanced Raman spectroscopy (SERS) is a type of molecular vibrational spectroscopy with high selectivity and sensitivity. SERS could provide fingerprints information of analytes in a label-free, non-destructive, instant, and simple way. Since the first discovery of enhanced Raman signals of pyridine from roughened silver surfaces in the 1970s, SERS has been widely applied in analytical areas such as food safety [1], environmental detection [2], disease diagnosis [3], pesticide residues [4], and biochemistry [5]. The common application for SERS is direct detection of the target analyte near or adsorbed on the plasmonic-enhanced substrate. The development of nanotechnology has led to an expansion of SERS applications by using metallic nanomaterials as active substrates. Metallic nanomaterials, especially those that are part of the noble metals, can provide excellent SERS sensitivity even down to the single molecule level. In the real world, the target analytes usually exist in complex systems, and it is difficult to obtain information on analytes from complex systems by the SERS method.

It is necessary to develop an advanced analytical method, combined with a separation technique such as extraction or chromatography [6]. Chromatography is a commonly used separation technique for detecting analytes in mixture samples. Examples are high performance liquid chromatography-mass spectrometry (HPLC-MS) [7], high performance liquid chromatography-ultraviolet (HPLC-UV) [8], and gas chromatography-mass spectrometry (GC-MS) [9]. These combined analytical methods are reliable. However, the GC and HPLC are expensive, time consuming, and complex, which is not convenient for on-site detection [10]. Thin-layer chromatography (TLC) is a simple and instant separation method, which has been commonly used as a cost-effective technique for separating and identifying products in organic synthesis. The tandem of TLC and SERS (TLC-SERS) is an efficient technology for separating and identifying analytes from complex systems. The TLC-SERS has been employed in food safety [11], healthcare [12], chemical analysis [13], and environment protection [14].

In this review, we will focus on the research of the application of TLC-SERS. First, we will briefly overview the phenomenon of SERS and the basic concept of the TLC method. Next, we will discuss the construction and application of TLC-SERS, which show excellent performance in detecting analytes from complex samples. Finally, we will conclude with the challenges and prospects of TLC-SERS, as well, such as how to improve the sensitivity, uniformity, and separation efficiency of TLC-SERS. Meanwhile, it can be predicted that TLC-SERS technology will have broad development space in terms of analysis and instrumentation. Therefore, it is particularly important to review the progress of TLC-SERS.

## 2. SERS

SERS is a powerful analytical technique that has received considerable attention [15]. In the 1970s, the intense Raman spectra of pyridine was observed from the rough surface of a silver electrode [16]. In 1977, Van Duyne et al. found a new discovery through electrochemical experiments [17,18]. The Raman signal of pyridine adsorbed on the rough silver surface showed higher intensity compared with ordinary Raman spectroscopy, and the increase was nearly 10^5^–10^6^ times. The significant enhancement of Raman signal was highly related to the feature of noble metal substrate [19,20,21,22,23,24,25]. Additionally, the SERS phenomenon was observed from transition metals and semiconductor materials. The detection performance of SERS is highly dependent on the enhancement substrate [26]. Thus, numerous nanomaterials were developed and used as SERS substrates. There are many strategies for preparing SERS substrates [27,28,29,30,31,32] such as chemical synthesis, photolithography, and self-assembly. The colloidal nanomaterials is one of the most commonly used substrates in SERS [33]. The nanoparticles with different geometry such as nanorods, nanocubes, and nanostars have shown high performance in SERS sensing. However, in the real world, the analytes mostly exist in complex systems, and the interference from other components hinder the detection performance of SERS. Therefore, the sample pretreatment or separation process was necessary in tandem with SERS [34,35].

## 3. TLC

TLC is an effective separation technology and has shown excellent performance in isolating analytes from mixture samples. The mixture sample was firstly spotted onto the TLC plate, then the TLC plate was placed in a chromatographic chamber with a suitable mobile phase. Next, the analytes migrated with the development of the mobile phase. After that, the TLC plate was illuminated under the UV light or sprayed with a color reagent. Different components undergo adsorption–desorption equilibrium during the development process. The adsorption capacity of the mixture is proportional to the polarity. The components with strong adsorption move slowly with the developing agent, while the components with weak adsorption move faster. The position of the analytes on the TLC plate is represented by the ratio shift value (Rf), and the components in the mixture are separated due to their different inherent polarities [36]. TLC has the advantages of low cost, fast separation speed, simple process, and low requirements for sample pretreatment in detection.

The stationary phase of the TLC plate is a key factor in the separation process. In the conventional TLC process, the silica gel, alumina, diatomaceous earth, and cellulose were used as the stationary phase. The mobile phase is divided into single- and multi-solvent according to the target mixture. Ewa Bębenek et al. studied the lipophilicity parameters of birch resin and betulin ester derivatives on silica TLC plates [37], and the mobile phase comprised the mixture of acetone and tris buffer. High-performance thin-layer chromatography (HPTLC) was proposed to improve the separation efficiency, in which the particles with small diameter and narrow size distribution were used as the stationary phase. The smaller size of particles in the stationary phase could provide a high theoretical plate height (H) and, hence, higher separation efficiency. HPTLC has been widely used in quantitative analysis in various fields. Oellig et.al. first separated ricinoleic acid from rye by using HPTLC on a silica-gel plate [38], in which the mixture solution of cyclohexane/diisopropyl ether/formic acid (86:14:1) was chosen as the mobile phase. After separation, the ricinoleic acid spot on the HPTLC plate was visualized by UV light (254 nm) irradiation, the limit of detection could achieve 0.1 ppm. In order to obtain the quantitative and qualitative information in a more reliable and sensitive way, the analytical instrument was tandem with TLC, such as the TLC-Fourier transform–infrared microscopy [39], TLC-mass spectroscopy [40,41,42] and TLC-Raman spectroscopy [43].

## 4. TLC-SERS

SERS spectroscopy is a facile and powerful analytical method, which could provide the inherent molecular information of analytes. The combination of chromatography and SERS has presented obvious advantages in separating and identifying analytes from mixture samples [44,45,46,47]. The simple, instant, and cost-effective merits of TLC-SERS make it widely applicable in analytical and organic chemistry [48]. Zhang et al. used a simple TLC-SERS analysis technique to effectively in situ monitor the chemical reaction process as shown in Figure 1 [49]. Since the pioneering research of TLC-SERS was developed by Zeiss and his coworkers [50], this technology has been successfully applied for separating and detecting analytes in complex samples. The interference from the molecules of the mobile phase is negligible in TLC-SERS. First, the mobile phase used in TLC is commonly an organic solvent, and after separation, the TLC plate is dried in air or heat conditions; thus, the molecules of the mobile phase are evaporated as their volatile property [51,52]. Second, the control experiment is usually developed in the TLC-SERS method, and no SERS signal of molecules of the mobile phase is measured from the TLC plate [53,54]. Following the understanding of the SERS and TLC, we will discuss the application of TLC-SERS in real cases, which include the detection of environmental pollutants [55], pesticides, food additives [12], food spoilage, biological sample [56], and chemical substances. Thereby, we hope TLC-SERS could be effectively applied to detect analytes from practical samples.

### 4.1. Environmental Pollutants

The environment is one of the most important issues for human beings. With the development of industry and economy, environmental pollution has occurred and presents a serious trend. There are several kinds of pollutants commonly presented in the environment such as organic pollutants, heavy metal pollutants, inorganic pollutants, biological, and radioactive pollutants. Inorganic, radioactive, and biological pollutants have relative single sources and are easily identified. Organic pollutants have strong toxicity and difficult degradation features. With a wide range of characteristics, once the pollutants are enriched for a long time, this will seriously threaten the ecological environment and human health. Therefore, it is urgent to develop analytical technologies that can detect organic pollutants in a simple and rapid way. There are several methods commonly used for detecting organic pollutants including gas chromatography [57], FTIR spectroscopy [58], and electrochemical analysis [59]. These methods are reliable but the complicated sample pretreatment hinders the wide application. Li et al. proposed TLC-SERS technology for on-site detection of benzene pollutants in water as shown in Figure 2 [14], in which the aggregating agent, concentration of Ag colloids, integration time, and laser power were optimized. The qualitative and quantitative detection of p-toluidine, p-nitroaniline, and m-phenylenediamine from mixture samples was achieved, and the detection accuracy was comparable with the GC-MS method. No sample pretreatment was needed before the TLC-SERS method, and the detection process took a short time. Thereby, errors caused by transportation and storage were reduced. It provides a fast and convenient way for on-site monitoring of environmental pollutants. Takei’s group prepared a TLC plate with a built-in SERS layer [11], in which the Au layer (40 nm) was firstly deposited onto the surface of the quasi-monodisperse silica nanoparticles by vacuum evaporation. The mixture of 1,2-bis(4-pyridyl)ethylene (BPE), crystal violet (CV), and rhodamine 6G (R6G) was successfully separated and detected. The built-in SERS layer TLC is different with normal TLC-SERS whereby plasmonic colloids are added after separation of the sample. Incidentally, having a built-in enhanced substrate significantly facilitates detection and provides better uniformity of SERS signals. This method provides a fast and convenient route for on-site monitoring of environmental pollutants. Zhang et. al. demonstrated an advanced strategy to prepare the TLC plate with metal-organic frameworks (MOFs), and the gold nanoparticles were composed in the MOF layer to form an eTLC-SERS device [60], which was used for detecting R6G with outstanding sensitivity. Compared with the normal TLC-SERS method, the eTLC-SERS device showed several merits. First, it eliminates interference from the additional nanoparticles and has a simpler process. Second, the eTLC platform could provide sensitive instant SERS sensing with excellent uniformity. The eTLC-SERS method based on the MOF layer can shorten analytical time and be used in a wider range of applications.

The TLC-SERS technology has been continuously improved in the detection of environmental pollutants, from the optimization of experimental conditions to new materials used to fabricate the TLC plate. Guaranteeing the feasibility of TLC-SERS for detecting more types of pollutants in the environment is one direction of this technology. Moreover, the miniaturization and simplification of the instrument would also be beneficial in detecting pollutants in the environment by TLC-SERS.

### 4.2. Illegal Additives

Recently, food safety issues have created considerable concerns as foodborne diseases are highly related with the morbidity in the world. Nearly one third of people in industrialized countries suffer from foodborne illness every year. Food additives were commonly used in the modern food industry as they could provide brilliant color, rich flavor, or long shelf life. Meanwhile, chemicals are illegally added in food stuffs for imparting special standards or effect, which brings serious problems to food safety. Thus, instant, accurate, and simple methods for detecting food additives is of high significance to ensure food safety. Many technologies have been used to detect illegal additives in food; most of them involve the application of chromatography coupled with different detectors [61,62,63]. TLC-SERS technology has been widely applied for separating and detecting harmful ingredients in food.

Botanical dietary supplements (BDS) have shown positive effects on chronic or systematic diseases from long-term ingesting. Lu’s group applied the TLC-SERS method for rapid on-site detecting adulteration of antidiabetes drugs in botanical dietary supplements [12], in which the accuracy was determined by liquid chromatography–triple quadrupole mass spectrometry. As adding ephedrine analogues into BDS, which brought difficulty in detecting adulteration of BDS. Lv et al. applied the TLC-SERS method for detecting ephedrine and its analogues in BDS [64], in which the characteristic peaks of additives were firstly measured. The supervised PLS-DA method was used to clarify the differences of the four ephedrine analogues. The detection method was effective for detection of BDS adulterated with ephedrine analogues, and the novel TLC-SERS mode could ensure the quality of BDS in a simple and instant way. The enhanced substrate used in most TLC-SERS was hydrophilic, which hinders the sensitive detection of hydrophobic analytes as their incompatibility with water. Zhu et al. synthesized a universal silver colloid, which is applicable for both hydrophilic and hydrophobic analyte sensing by TLC-SERS [65]. After optimizing the preparation conditions of the silver colloid, they successfully detected hydrophilic and hydrophobic adulterants in real BDS samples by TLC-SERS.

The performance of TLC-SERS is highly related with the chromatographic materials of the TLC plate. Currently, most TLC-SERS were based on a silica gel or cellulose TLC plate. Several strategies were proposed for fabricating the TLC plate with a novel stationary phase. Gao et al. [66] developed molecularly imprinted polymer TLC-SERS (MIPs-TLC-SERS) to detect Sudan red in pepper powder, and the limit of detection reached 1 ppm. The pretreatment of the pepper sample was nearly eliminated and the separation process was extremely instant due to the “lock and key” principle between MIPs and Sudan I. Kong et al. have fabricated a diatomite TLC plate; the periodic pores on diatomite have photonic crystal features that could bring additional SERS enhancement [67]. The TLC-SERS method based on the diatomite plate successfully separates and detects Sudan I from real chili product as shown in Figure 3. Zhao and coworkers used the silver nanorod arrays as the stationary phase for constructing the ultra-thin-layer chromatography-SERS (UTLC-SERS) method, in which the nanorod arrays were prepared by oblique angle deposition (OAD) [68,69]. The novel plate was used for the detection of PAHs from cooking oil by TLC-SERS, the ultra-thin-layer stationary of silver nanorod arrays provided uniform SERS signals and with little amounts of sample consuming. The TLC-SERS method can also be applied for detecting poppy peels in hot pot condiments, melamine from milk, and hydrophilic vitamins from food [45,70,71,72]. The TLC-SERS method is preferable for on-site detection of additives in food to enhance food safety.

### 4.3. Pesticide Residues

In modern agriculture, pesticides play an important role as the chemical control method, but their residues on the surfaces of vegetables and fruits are difficult to be avoided totally. TLC-SERS has been applied for detecting pesticide residues in food samples. Our group used TLC-SERS for separating and detecting carbendazim in orange juice and kale, in which the diatomite chip exhibited excellent separation and detection performance [73]. The detection could be finished in 5 min, and the pyrimethanil, pymetrozine, and carbendazim could be detected simultaneously by the TLC-SERS method as shown in Figure 4. TLC-SERS was also used to separate and detect organophosphate pesticides from tea leaves [74]. In that research, the different TLC plate, Au/Ag colloid, and concentration of enhanced substrate were investigated; five different organophosphorus pesticides were identified with a limit of detection down to 0.1 ppm. The enhanced substrate is a critical issue that is associated with the performance of TLC-SERS. Metallic colloids are commonly used as enhanced substrates in TLC-SERS, and that enhancement effect usually varied from one substrate to another, and from one spot to another of the same TLC plate. Kang and coworkers fabricated dendritic-like gold nanomaterials and wrapped them with carbon fiber [54]; this novel substrate could provide a huge number of hotspots for SERS as the dendritic-like nanostructure. That material was successfully used as an enhanced substrate in TLC-SERS for detecting acetamiprid pesticides in cabbage. The needle tip of the SERS substrate was inserted into the stationary phase of the TLC plate to collect the molecule information of analytes. Recently, the TLC-SERS method with high sensitivity was achieved by using a state translation process of metallic colloid from wet state to dry state, namely thin-layer chromatography-dynamic metastable state nanoparticle-enhanced Raman spectroscopy (TLC-MSNERS) [75]. The metallic nanoparticles tended to move closer to produce hot spots during the solvent volatilization process. Additionally, it is a challenge to seize the state of metallic colloid before the solvent is completely dry to collect the best SERS signal. Du’s group prepared Ag nanoparticles by using amphiphilic polymer polyurethane as the stabilizing agent, in which the polyurethane could form micelle to adsorb metallic nanoparticles and analytes. The polymer-Ag nanocomposite was used as an enhanced substrate in the TLC-MSNERS method to successfully separate and detect the mixture pesticides of triazophos, phosmet, and thiabendazole from fruit [76]. The time-dependent SERS signal showed that the polymer could significantly improve the stability of metallic colloid during their volatilization process.

### 4.4. Toxic Ingredients

Corrosion and deterioration problems commonly exist in food stuff because the improper storage of food leads to infection with bacteria. In seafood, the bacteria could convert histidine into histamine, which is associated with pathological processes, such as acute allergies and the inflammation of the immune system. Our group applied TLC-SERS for separating and identifying histamine from real spoiled tuna [53]; the diatomite photonic crystal was used to construct the TLC plate. The spoiled tuna sample was directly applied on the diatomite TLC plate without pretreatment; the separation process was within 3 min. After separation, the molecule information of histamine was measured by SERS, and the concentration of histamine in the decomposed tuna was almost at 150 ppm. Histamine has a low Raman cross-section and colorless features, which hinder the sensitive detection by normal TLC-SERS. Derivatized TLC-SERS methods were developed, in which the fluorescamine was employed for derivatizing histamine [77,78]. As shown in Figure 5, after derivatization, the new compound presented a fluorescence feature and intense Raman signal, and the limit of detection of histamine from mixture sample by TLC-SERS could achieve 9 ppb. Another challenge in the TLC-SERS method for food spoilage detection is in the nonlinear relationship between the SERS signal and the concentration of analytes, which brings difficulty in quantitative detection. The machine learning analysis based on support vector regression and principal component analysis was introduced in TLC-SERS; the quantitative model achieved excellent predictive performance for monitoring the spoiled process of the tuna sample in 48 h [79]. This study indicates that the TLC-SERS combined with machine-learning analysis is a simple, reliable, and accurate method for on-site detection and quantification of toxic components in food.

Improper storage of nuts or soybeans can produce aflatoxins (AFs), which bring irreversible damage to the human body. Qu et al. established a portable, fast, and simple method for on-site detecting of AFs in peanuts by TLC-SERS [52], in which the concentrated gold colloid was used as the SERS substrate. This method showed high selectivity and sensitivity for identifying four kinds of AFs in complex samples from moldy peanuts. The result proves that TLC-SERS could be effectively applied for distinguishing four AFs, which shows good prospects for on-site qualitative monitoring toxic materials in food. The citrus flavonoids, benzidine, and 4-aminobiphenyl were also successfully detected from food stuff by TLC-SERS.

### 4.5. Biological Molecules

TLC-SERS has been used to detect biological molecules. Apomorphine is a type of short-acting dopamine agonist for treating Parkinson disease. The instant detection of apomorphine plays an important role in human health. Lucotti et al. developed a method for detecting apomorphine in blood plasma by TLC-SERS. The detection process could be finished in a few minutes, in which the information of drugs was observed from the SERS spectra after TLC separation. Furthermore, they proved the interaction between the analyte and metallic SERS substrate by density functional theory calculation [80].

Kong et. al. have fabricated plasmonic nanoparticles-decorated diatomite biosilica for separating and detecting analytes in blood plasma [48]. That device demonstrated significant potential in biomedical diagnosis; the phenethylamine and miR21cDNA were separated and identified in human plasma. The experimental results showed high sensitivity, which is more than 10 times compared to the commercial silica-gel TLC plate. In order to further improve the sensitivity in TLC-SERS, a microfluidic diatomite analytical device (μDADs) was developed [81]. The μDADs device was prepared by spin-coating and tape-stripping the diatomite channel with width and height at 400 and 30 μm, respectively. The μDADs device showed high confinement of the analyte due to the ultra-small dimension of the diatomite channels, which demonstrated high sensitivity for sensing cocaine (10 ppb) in human plasma. Sivashanmugan et al. applied TLC-SERS for monitoring cannabis-based drug abuse; a trace number of cannabis biomarkers was successfully detected in biofluids, and the multivariate statistical method of PCA was used to quantitatively evaluate the concentration of tetrahydrocannabinol and its metabolites [82]. Such a portable sensing platform can play a pivotal role in future forensic and biological applications. TLC-SERS has been used for detecting nicotine metabolites and paracetamol in urine samples [83,84]. The simple and instant features of TLC-SERS enable the potential application in bioanalysis.

### 4.6. Chemical Substances

TLC has advance features such as simple operation and instant process, which plays an important role in differentiating chemicals and monitoring synthetic organic reactions. The limited specificity and low sensitivity hinder the application of TLC in identifying chemicals. SERS could provide specific information of target molecules with high sensitivity. Ian White’s group has performed advance work on TLC-SERS based on plasmonic paper substrates [85,86,87], in which the plasmonic Ag NPs were firstly deposited onto the surface of the filter paper by inkjet printing. The plasmonic paper substrates provide a high SERS enhancement effect as the dense Ag NPs. In addition, the inherent porous structure of cellulose fiber in the paper provides unique capabilities, which function as the TLC plate. The paper-based TLC-SERS methods have successfully separated and detected different pigments from mixture. Van Duyne’s group has applied TLC-SERS for detecting artist dyestuffs [88]. The low sensitivity of TLC was overcome by combing it with SERS, and the colorants such as alizarin, purpurin, carminic acid, lac dye, crocin, and Cape jasmine were successfully distinguished. Zhang et al. applied TLC-SERS for continuous and automatic on-site monitoring the chemical reaction processes [49]. After separated by TLC, the Raman signals provide all the information of the different components. The Raman mapping could cover a large area on the TLC plate, and the byproducts were identified by SERS spectra. This facile TLC-SERS method can be exploited in monitoring the progresses of organic reactions. The separation efficiency is a key factor in TLC-SERS for identifying chemicals. The normal silica-gel TLC plate is hard to separate chemicals with similar structure or polarity. Several types of materials such as silver/polymer nanocomposite, plasmonic diatomite, monolithic silica gel, silicon nanowires array, and silver nanorod array were used to fabricate the TLC plates, which could significantly improve the separation capability and signal uniformity of TLC-SERS [13,89,90,91,92]. These TLC plates were used for distinguishing polycyclic aromatic hydrocarbons, diterpenoic acids, malachite green isothiocyanate, 4-aminothiophenol, and dyes by TLC-SERS.

## 5. Conclusions and Future Perspectives

As a facile analytical method, TLC-SERS is simply combined separation technology (TLC) and Raman spectroscopy (SERS). There has been considerable work on applying the TLC-SERS method for either multiplex analyte detection or identifying harmful ingredients from food and environments. The TLC-SERS method is instant, simple, and cost-effective for on-site detecting of analytes in complex systems. Nevertheless, opportunities and challenges remain in TLC-SERS, for example, to improve the separation efficiency, detection sensitivity and reproducibility in application. Researchers should fabricate high-performance TLC plates, optimize SERS substrates, and precisely control the distribution of enhanced substrates on the TLC plate.

Most research works have attempted to show proof-of concept about the TLC-SERS method as a facile analytical method. Despite possible impediments such as sensitivity, reproducibility, and separation efficiency issues, TLC-SERS has the potential to promote the field of analytical science. It also has some valuable research directions. In the future, research about TLC-SERS should also focus on moving the application from lab to on-site. Therefore, first, the portable Raman spectrometer is a future direction that would be beneficial for on-site sensing by TLC-SERS. Second, the engineering plasmonic-based stationary phase in the TLC plate construction could improve the detection sensitivity and reproducibility. Third, the separation efficiency could be improved by using porous materials or polymers. For example, the smaller size of particles in the stationary phase could provide a lower theoretical plate height and provide higher separation efficiency. Additionally, by taking advantage of the chemometric analysis or machine learning, more accurate and quantitative information could be obtained from the signal. All in all, we believe that this review may help many researchers’ work on analytical chemistry, environment protection, food safety, and biomedicine.

## Figures and Tables

**Figure 1 biosensors-12-00937-f001:**
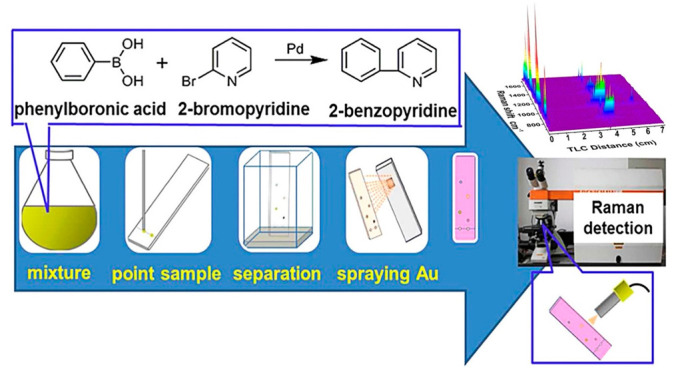
On-site quantitative monitoring of organic reaction by TLC-SERS. Reprinted with permission from ref. [49]; Copyright 2014 American Chemical Society.

**Figure 2 biosensors-12-00937-f002:**
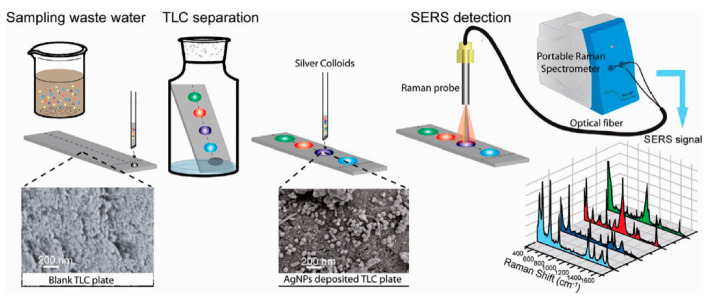
On-site detecting of organic pollutants from wastewater by TLC-SERS. Reprinted with permission from ref. [14]; Copyright 2011 American Chemical Society.

**Figure 3 biosensors-12-00937-f003:**
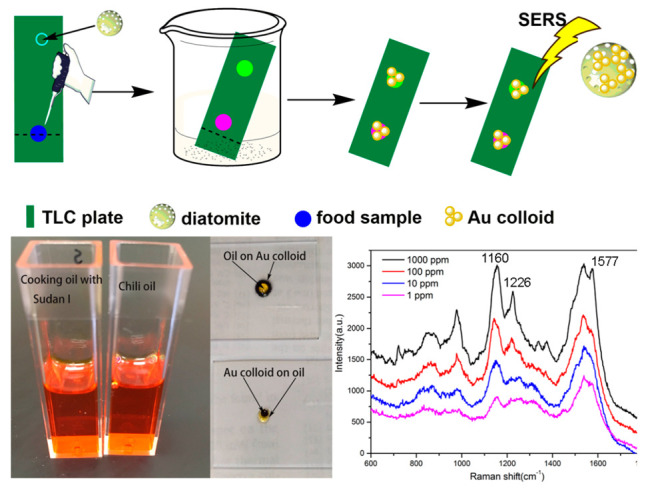
Schematic representation of the TLC-SERS method by diatomite plate and the application for detecting Sudan dye in chili oil. Reprinted with permission from ref. [67]; Copyright 2017 Elsevier.

**Figure 4 biosensors-12-00937-f004:**
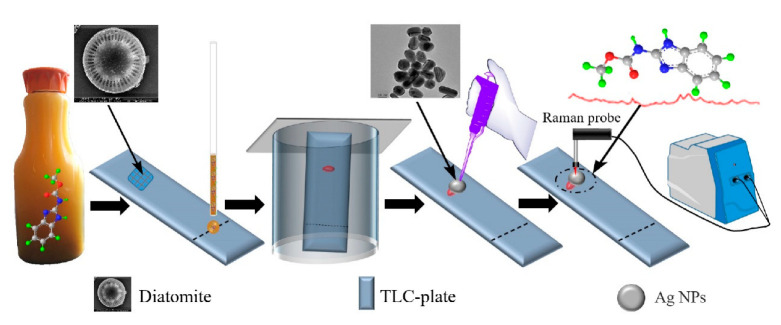
TLC-SERS detection of pesticide residue from food sample. Reprinted with permission from ref. [73]; Copyright 2021 Elsevier.

**Figure 5 biosensors-12-00937-f005:**
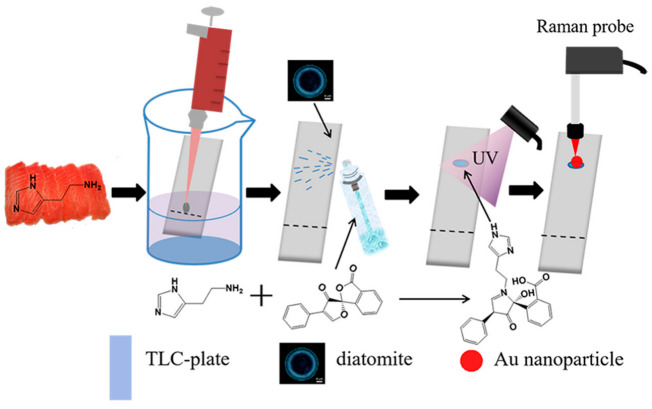
On-site derivatized-TLC-SERS for detecting histamine in tuna. Reprinted with permission from ref. [77]; Copyright 2022 Elsevier.

## Data Availability

The data presented in this study are available on request from the corresponding author.

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
