# Peer review of "Review of Thin-Layer Chromatography Tandem with Surface-Enhanced Raman Spectroscopy for Detection of Analytes in Mixture Samples"

_biosensors, 2022, doi:10.3390/bios12110937_

Round 1

Reviewer 1 Report

Report on Manuscript ID biosensors-1968987:  Review of thin layer chromatography tandem with surface-enhanced Raman spectroscopy for detection of analytes from mixture sample Special Issue: Nanostructures for Tip and Surface Enhanced Vibrational Spectroscopy (TERS, SERS, SEIRA, SECARS). By Meizhen Zhanga, Qian Yua, Jiaqi Guob, Bo Wuc and Xianming Kong.

These type of Review articles are always welcomed by the interested scientists working in the use of the same or adaptions of a given methodology, and hence, the present manuscript provides very useful information for all applied scientists who take advantage of SERS to monitor the presence, at very low concentrations of some molecule in study. As the authors mention in the introductory remarks, thin layer chromatography (TLC) on the other side, offers a cheap, easy to use technique to separate target molecules inside a complex medium, where the analyte of interest is existing in very mixed conditions.

The review of applications to different subjects of interest in modern lives I find them interesting and useful.

My only commentary to the authors is that I don´t find a discussion on the possible phenomenon of interference in the SERS signal expected from the target molecule in study, by a possible expected SERS signal produced by the molecules of the mobile phase. Do these produce a background or a collaborative signal when adsorbed by the sprayed nanoparticles on the target spot of the TLC plate? What is the experience from the works reviewed who supply the information for this article? I expected to see a discussion about this concern, in the main text of this article.

Author Response

Dear Reviewer
We appreciate the valuable comments and suggestions from you. We have carefully addressed these concerns and included detailed response in this letter.

The manuscript "Facile synthesis of diatomite/β-cyclodextrin composite and application for the adsorption of diphenolic acid from wastewater" is well written. This can be considered worth publishing in journal "Materials" after following modifications: These type of Review articles are always welcomed by the interested scientists working in the use of the same or adaptions of a given methodology, and hence, the present manuscript provides very useful information for all applied scientists who take advantage of SERS to monitor the presence, at very low concentrations of some molecule in study. As the authors mention in the introductory remarks, thin layer chromatography (TLC) on the other side, offers a cheap, easy to use technique to separate target molecules inside a complex medium, where the analyte of interest is existing in very mixed conditions.

The review of applications to different subjects of interest in modern lives I find them interesting and useful.

My only commentary to the authors is that I don´t find a discussion on the possible phenomenon of interference in the SERS signal expected from the target molecule in study, by a possible expected SERS signal produced by the molecules of the mobile phase. Do these produce a background or a collaborative signal when adsorbed by the sprayed nanoparticles on the target spot of the TLC plate? What is the experience from the works reviewed who supply the information for this article? I expected to see a discussion about this concern, in the main text of this article.

Answer: Thanks for the comments and valuable suggestions. ‘The interference from the molecules of the mobile phase is negligible in TLC-SERS. First, the mobile phase used in TLC are commonly organic solvent, after separation the TLC plate was dried in air or heat conditions, thus the molecules of the mobile phase are evaporated as their volatile property[51, 52]. Second, the control experiment usually developed in TLC-SERS method, and no SERS signal of molecules of the mobile phase measured from the TLC plate[53, 54].’

Reviewer 2 Report

The article is a review of TLC and SERS, and the authors have covered the major developments in the field.  Need to edit English language.

Author Response

Dear Reviewer
We appreciate the valuable comments and suggestions from you. We have carefully addressed these concerns and included detailed response in this letter.

Thanks for the valuable suggestions. We have checked the English language in the manuscript carefully.

Reviewer 3 Report

In the present review authors have reviewed combined TLC-SERS  technology for multiplex detection of analytes for identification of hazardous compounds from biological samples, food, and the environment. The TLC-SERS method have several advantages like rapid, simple and cheaper analysis as POC settings.

I have few comments--the need to provide more insights and in-depth analysis of recent publications. Additionally, authors should include recent most studies. There are several references which are more than 10 years old. I think authors should latest studies and provide future directions, instead of mere summary.

Author Response

Dear Reviewer
We appreciate the valuable comments and suggestions from you. We have carefully addressed these concerns and included detailed response in this letter.

In the present review authors have reviewed combined TLC-SERS technology for multiplex detection of analytes for identification of hazardous compounds from biological samples, food, and the environment. The TLC-SERS method have several advantages like rapid, simple and cheaper analysis as POC settings.

I have few comments--the need to provide more insights and in-depth analysis of recent publications. Additionally, authors should include recent most studies. There are several references which are more than 10 years old. I think authors should latest studies and provide future directions, instead of mere summary.

Answer: Thanks for the comments and valuable suggestions. More discussions of the recent publications were added in the manuscript. More recent publications were cited in the text.

  1. The Journal of Physical Chemistry C 2019, 123, (40), 24714-24722.
  2. The Journal of Physical Chemistry Letters 2020, 11, (9), 3573-3581.
  3. Spectrochimica Acta Part A: Molecular and Biomolecular Spectroscopy 2020, 118589.
  4. Sens Actuators B Chem 2022, 357.
  5. Angewandte Chemie International Edition 2019, 58, (41), 14452-14456.
  6. Food chemistry 2019, 287, 363-368.
  7. J Pharm Biomed Anal 2019, 174, 340-347.
  8. Food Bioscience 2022, 49.
  9. Spectrochim Acta A Mol Biomol Spectrosc 2022, 280, 121464.

The future direction of TLC-SERS was listed in the manuscript.

Round 2

Reviewer 3 Report

No additional comments.